# Development of a Novel Primer–TaqMan Probe Set for Diagnosis and Quantification of *Meloidogyne enterolobii* in Soil Using qPCR and Droplet Digital PCR Assays

**DOI:** 10.3390/ijms231911185

**Published:** 2022-09-23

**Authors:** Yuan Chen, Haibo Long, Tuizi Feng, Yueling Pei, Yanfang Sun, Xinchun Zhang

**Affiliations:** 1Key Laboratory of Integrated Pest Management on Tropical Crops, Ministry of Agriculture and Rural Affairs, Environment and Plant Protection Institute, Chinese Academy of Tropical Agricultural Sciences, Haikou 571101, China; 2Hainan Key Laboratory for Monitoring and Control of Tropical Agricultural Pests, Institute of Environment and Plant Protection, Chinese Academy of Tropical Agricultural Sciences, Haikou 571101, China

**Keywords:** *Meloidogyne enterolobii*, TaqMan probe, qPCR, ddPCR, identification, quantification

## Abstract

Early detection of pathogens before the planting season is valuable to forecast disease occurrence. Therefore, rapid and reliable diagnostic approaches are urgently needed, especially for one of the most aggressive root knot nematodes, *Meloidogyne enterolobii*. In this study, we developed a novel primer–TaqMan probe set aimed at *M. enterolobii*. The primer–probe set was successfully applied in the identification and quantification of *M. enterolobii* via qPCR technology. It was also suitable for improved PCR technology, known as ddPCR analyses, and this work presents the first application of this technology for plant parasitic nematodes. Compared with qPCR, ddPCR exhibited better performance with regard to analytical sensitivity, which can provide a more accurate detection of *M. enterolobii* concealed in field soil. In addition, we generated standard curves to calculate the number of eggs in soil using the qPCR and ddPCR platforms. Hopefully, the results herein will be helpful for forecasting disease severity of *M. enterolobii* infection and adopting effective management strategies.

## 1. Introduction

Plant parasitic nematodes (PPNs) are considered to be a huge threat to agricultural crops across the world, estimated to cause over $100 billion USD of loss annually, with about 500 million USD spent on PPN management per year [1]. Root knot nematodes (RKNs; *Meloidogyne* spp.) represent one of the economically important PPNs. RKNs are obligate sedentary endoparasites that require nutrients from host plants to complete their life cycle, including eggs, four juvenile stages, and the adult male or female. The adult females lay eggs near or at the surface of the root and form a gelatinous egg mass which contains up to 1000 eggs. The infective second-stage juveniles (J2s) hatched from the eggs invade the plant root tip and migrate in the vascular system. Then, RKNs become sedentary at the chosen feeding sites and induce the formation of 5–7 giant cells in host plants to provide nutrients for RKN growth and reproduction. After three molts at the feeding site, they develop into adult females or males. The females exude egg masses within the gall tissue of the host roots to cause repeated infections in one planting season, or directly into the rhizosphere which will stay inactive until the host crops are planted in a new growing season [2,3,4]. Among the RKNs, *M. enterolobii* is an emerging threat to crop production and has received more attention since it was first reported in 1983 in Hainan, China [5]. To date, a broad range of cultivated crops in subtropical and tropical areas were reported to be prone to attack by *M. enterolobii*, even plants carrying resistance genes. Due to similar symptoms such as yellowing, stunting, and large galls, it is hard to distinguish the infection of *M. enterolobii* from other common *Meloidogyne* species in the field, such as *M. incognita* and *M. javanica* [6].

As one of the most destructive soilborne pathogens residing in roots, accurate identification and quantification in soil are significant to assess the potential threats of *M. enterolobii*. For identification of *Meloidogyne* spp., morphology-based analyses using light microscopy or autofluorescence and a series of molecular methods based on DNA- and/or protein information have been developed [7]. *M. enterolobii* was originally identified by the perineal pattern of females, with characteristics of an oval shape, dorsal arch usually high and round, weak lateral lines occasionally present, large phasmids with occasional breaks of striation laterally, and a circular tail tip area lacking striae [5]. In addition, *M. enterolobii* can be distinguished by the morphometrics of J2s and males, such as stylet length, height and width of stylet knobs, and neck length [8]. However, it is easy to confuse *M. enterolobii* with other RKNs with similar morphological features or overlapping dimensions. Isozyme phenotyping is a biochemical-based diagnostic technique for detection of the young adult females of *M*. *enterolobii*. Enzyme phenotypes, specifically esterase (EST), malate dehydrogenase (MDH), superoxide dismutase (SOD), and glutamate–oxaloacetate transaminase (GOT), have been used to characterize different species of *Meloidogyne* spp. On the basis of the protein sequence of *M. enterolobii*, it can be detected using the unique pattern of two distinct esterase bands and one malate dehydrogenase band [7,9].

DNA-based molecular methods are now the most prevalent diagnosis tools for nematodes. In recent years, molecular diagnostics targeting different regions of the genome and mitochondrial DNA have been developed for *M*. *enterolobii* identification. Restriction fragment length polymorphism (RFLP), random amplified polymorphism DNA (RAPD), amplified fragment length polymorphism (AFLP), and inter-simple sequence repeat (ISRR) analyses were successfully applied to detect *M. enterolobii* [10,11,12,13,14]. In addition, various PCR primer sets were designed by amplifying species-specific fragments of mitochondrial DNA regions [15,16], the intergenic region from the ribosomal DNA [12,17], the satellite DNA [6,18], and a sequence-characterized amplified region (SCAR) of genomic DNA [19]. PCR-based diagnostics possesses the advantages of being fast and sensitive, whereas only high-quality DNA templates extracted from single juvenile or adult nematodes are applicable, such as RFLP, RAPD, AFLP, and ISRR methods. Although the loop-mediated isothermal amplification (LAMP) assay was successfully verified to identify the crude DNA isolated from root and soil samples [20], it still cannot remedy the inability to accurately quantify of *M. enterolobii* in soil analyzed using all the methods mentioned above.

A hydrolysis probe, known commercially as a TaqMan probe, is a fluorogenic single-stranded oligonucleotide probe that binds only the DNA sequence between two PCR primers [21]. It can indirectly monitor the amount of target amplicon as only a specific PCR product generates a fluorescent signal in TaqMan PCR [22]. Quantitative real-time PCR (qPCR) using a TaqMan probe was developed to identify and quantify *Meloidogyne* spp. in soil and root galls, such as *M. chitwoodi*, *M. fallax*, *M. minor*, and *M. hapla* [23,24,25]. For *M. enterolobii*, two locked nucleic acid (LNA) probes were developed and validated for fast and reliable detection and identification of DNA templates extracted from J2s hatched from soil and root samples [26]. Furthermore qPCR, the TaqMan probe can be equally applied to droplet-digital PCR (ddPCR), which is a third generation of PCR that can clonally amplify and directly quantify nucleic acid targets within a sample [27]. As ddPCR technology generally gives higher sensitivity, better accuracy, and more stable replications, it has been widely used to detect mutations, quantify specific nucleic acid species, and analyze copy number variations of specific genes [28]. ddPCR has successfully been brought into clinical use for infectious disease diagnosis. For plant pathogens, ddPCR has been developed for the diagnosis of bacteria, fungi, viruses, and phytoplasma, but unfortunately not of PPNs [29,30].

Here, our aim was to develop a primer–TaqMan probe set suitable for qPCR and ddPCR methods to identify and quantify *M. enterolobii*. The specificity, accuracy, and sensitivity were assessed and compared between the two methods to detect *M. enterolobii* in field soil.

## 2. Results

### 2.1. Species-Specific Primers and TaqMan Probe Designing for M. enterolobii Identification

On the basis of the rDNA sequence alignment of common *Meloidogyne* species, we developed the specific primer–probe set within the ITS2 region to detect *M. enterolobii* (Figure 1). The selected primers were MEfwd (5′–GGGTCATTATCTTTCAAAGC–3′) and MErev (5′–TGATGATACATGCGAACA–3′), which yielded a PCR amplicon of 107 bp. The TaqMan probe (5′–ATTGCTTTTGTGGCTTCTTT–3′), named MEprobe, was synthesized and labeled with 6-carboxyfluorescein (FAM) on the 5′ end as the reporter dye and a nonfluorescent quencher with MGB ligands at the 3′ end. The Primer-Blast sequence alignment results showed that the primers/probe were only likely to target *Caenorhabditis elegans*, in addition to targeting *M. enterolobii*.

### 2.2. Optimization of Annealing Temperature for ddPCR Assay in M. enterolobii Identification

Following the manufacturer’s instructions, we confirmed that the final concentration of primers and probes at 0.90 μM and 0.25 μM, respectively, were suitable for detecting DNA templates of *M. enterolobii*. However, the annealing temperature needs to be optimized as it is crucial for reaction specificity. For this purpose, DNA templates of 5 ng of genomic DNA (gDNA) or total DNA extracted from 0.25 g of soil inoculated with 200 eggs of *M. enterolobii* were detected by ddPCR. As shown in Figure 2, the optimal annealing temperature was 53.8 °C for both two DNA templates, and this temperature was selected for subsequent ddPCR assays.

### 2.3. Primers and Probe Specificity Tests Analyzed by qPCR and ddPCR

To evaluate the specificity of designed primers (MEfwd/MErev) and probe (MEprobe) for *M. enterolobii*, various nematodes were selected to extract gDNA for qPCR and ddPCR applications. The detected nematodes included four phylogenetically related *Meloidogyne* spp., namely, *M. incognita*, *M. javanica*, *M. graminicola*, and *M. arenaria*, two species of other phytoparasitic nematodes (*Heterodera glycines* and *Rotylenchulus reniformis*), and one free-living species (nonparasitic nematode of *C. elegans*). In qPCR assays, only *M. enterolobii* exhibited fluorescence increases at a specific cycle ranging from 25.094 to 25.528 in three replicated wells, representing positive amplification of the target gene when the cutoff value was set to 38.500 (Figure 3A). In ddPCR assays, more than three positive droplets were considered to be a positive sample, which is the minimum acceptable value for the ddPCR Poisson precision calculation and to avoid false-positive calls [31]. According to this principle, we only observed high accumulation of positive droplets in the reaction containing gDNA of *M. enterolobii* (Figure 3B). Although sequence alignment results showed probable targeting on *C. elegans*, no amplification was achieved in qPCR or ddPCR. These results confirmed that the primer/probe set we developed showed high specificity for identification of *M. enterolobii*.

### 2.4. Quantitative Linearity to Quantify the gDNA Dilutions and Eggs in Soil of M. enterolobii by qPCR and ddPCR

To test the linearity and efficiency of qPCR and ddPCR techniques for identification of *M. enterolobii*, the diluted genomic DNA extracted directly from eggs and total DNA extracted from 0.25 g of soil containing 1–200 eggs was prepared for detection to draw standard curves. When analyzed using qPCR, the Ct values were plotted against the logarithm of gDNA quantities or the number of eggs in soil (Figure 4A,C). The slope of the standard curves was −3.4636 for gDNA dilutions tested using qPCR, equivalent to an average PCR efficiency of 94.4% (Figure 4A). Meanwhile, the slope of the standard curve was −3.4239 for twofold dilutions of total DNA extracted from soil containing 200 eggs, equivalent to an amplification efficiency of 95.9% (Figure 4C). In addition, standard curves for ddPCR were generated between the number of DNA copies·μL^−1^ and the gDNA quantities or the number of eggs in soil (Figure 4B,D). In all assays, both methods showed good linearity with coefficients of determination (*R*^2^) over 0.98, which was a strong metric of linearity.

To evaluate the repeatability of qPCR and ddPCR detection results obtained from triplicates of an assay, relative standard deviation (RSD) values were calculated for each DNA sample. For qPCR analysis, the RSD values for gDNA and total DNA from soil were in the range of 1.553–5.997% and 1.419–4.598%, respectively. For ddPCR technology, the RSD values varied from 0.605% to 10.728% for gDNA, and from 1.553% to 10.303% for total DNA from soil (Table 1). All the RSD values were below the acceptable criterion of 25% [32], revealing the good repeatability for qPCR and ddPCR to identify *M. enterolobii* using the designed primers and probe.

### 2.5. Comparison of the Minimum Detection Limit between qPCR and ddPCR Platforms

The limit of detection is the metric of sensitivity of analytical methods. To determine the minimum limits, two types of DNA samples were prepared and subjected to analyses. First, DNA templates extracted from one egg in 0.25 g of soil were diluted and converted into the quantity of eggs for detection (1/30, 1/150, 1/300, and 1/3000 nematode eggs). Second, tenfold serial dilutions were used to generate DNA standards with the concentrations of 1000, 100, 10, and 1 fg·μL^−1^. All samples were detected using qPCR and ddPCR separately. The results revealed that qPCR could detect 1/150 eggs in soil and 100 fg·μL^−1^ of gDNA, whereas ddPCR allowed for detection of 1/300 eggs in soil and 10 fg·μL^−1^ of gDNA (Table 2). Collectively, the ddPCR assay exhibited a lower detection limit than qPCR technology.

### 2.6. Assessments of Field Samples Quantified Using Shallow Dish, qPCR, and ddPCR Methods

To assess the utility of the primer–probe set in quantifying the egg density, 100 g of fresh soil samples were collected from Wenchang, Hainan, which were mainly infected by *M. enterolobii*. Three subsamples of 0.25 g were taken from the tested sample, and total DNA was extracted for qPCR and ddPCR analyses. The remaining soil samples were processed using the shallow dish method to collect the nematodes. After 3 days, about 434 juveniles hatched from soil were observed under a microscope, equivalent to 4.8 eggs per gram of dried soil. Meanwhile, the egg densities of *M. enterolobii* in field soil calculated using qPCR and ddPCR were 7.1 and 62.5 per gram of dried soil, respectively (Figure 5).

## 3. Discussion

As root knot nematodes are concealed in host roots and field soil, farmers may sometimes not realize that the crops are infected by RKNs until the end of the growing season when the root galls are observed in the harvested plants. In this aspect, early diagnosis of RKNs in field is necessary and financially beneficial at the beginning of the planting season. Traditional identification techniques based on morphological and anatomical observation require expert skills, and they are time-consuming and prone to human error [7]. Along with the increasing amount of genome sequencing data, molecular methods are employed for accurate, reliable, and rapid identification of RKNs. Species-specific molecular markers targeting the conserved regions of ribosomal DNA have been developed and applied to identification of *M. enterolobii* [15,16,17,18,19,20]. Although these methods are effective to differentiate *M. enterolobii* from other RKNs, they still have some drawbacks that cannot be overlooked. Firstly, these methods, such as RFLP, RAPD, AFLP, and ISRR, can only determine whether the target species exists in the tested sample or not, without accurate quantification. Secondly, these methods can only detect a single individual isolated from the tested sample, needing a complex and high-quality DNA extraction procedure. Lastly, most of these methods lack sensitivity in detecting low amounts of target DNA.

To overcome the drawbacks of existing identification methods, we designed a novel primer–TaqMan probe set within the ITS2 region of *M. enterolobii*. In our study, the primer–probe set was successfully used to identify *M. enterolobii* in a complex DNA background via qPCR assays. It was confirmed that the primer–probe set is highly species-specific, as only templates containing *M. enterolobii* DNA were detected. In previous study, two LNA probes 50^#^ and 17^#^ were designed to detect J2s of *M. enterolobii*, isolated and hatched from soil or root samples. Only one J2 in a background of 1000 untargeted nematodes was detectable using the LNA-based qPCR [26]. It is noteworthy that the TaqMan qPCR method could detect both pure DNA templates and total DNA directly extracted from field soil. Several studies have reported the use of TaqMan probe for detecting various *Meloidogyne* species when DNA was extracted from root galls or field soil [23,24,25]. In our tests, the pure DNA solution and total DNA extracted from soil containing eggs were both suitable for detecting *M. enterolobii* via qPCR. It showed high sensitivity with the minimum threshold of 100 fg·μL^−1^ of DNA solution and 1/150 eggs in 0.25 g of soil.

Since qPCR provides fast and high-throughput detection and quantification of target DNA fragments, it has been the leading tool for plant pathogen diagnostics. However, qPCR is easily affected by sample inhibitors, amplification efficiency, and the subjective cutoff values in practical application [33]. ddPCR is an improved, high-tech, and highly sensitive tool for diagnostics, which is broadly used in human disease detection [28]. It allows accurate calculation of absolute DNA quantities based on the number of positive and negative droplets observed without the need for external reference standards or controls [34]. In view of this feature, we introduced the ddPCR tool for identification and quantification of *M. enterolobii*, which was the first attempt in PPNs. To achieve a better separation of positive and negative droplets, the annealing temperature of ddPCR was optimized using a thermal gradient PCR. The best separation for both DNA targets, including pure DNA solution and total DNA extracted from soil containing eggs of *M. enterorobii*, was obtained at 53.8 °C, with the combination of a 0.90/0.25 μM primers/probe concentration. At this condition, we further ascertained that only samples containing *M. enterolobii* DNA templates generated several positive droplets, validating the primer–probe set specific for *M. enterolobii* in ddPCR assays. Then, the performance of ddPCR was compared with qPCR with regard to sensitivity, repeatability, and the detection limit. Both qPCR and ddPCR could detect the serial dilutions of pure DNA from 1 pg to 5 ng, with good linearity (*R*^2^ > 0.98) and reproducible quantification (RSD < 25%). In ddPCR assays, the lowest detectable DNA concentration was 10 fg·μL^−1^, representing a tenfold increase in analytical sensitivity compared with qPCR. Meanwhile, the detectable minimal number of eggs in 0.25 g of soil was 1/300, which was also more sensitive than qPCR. However, the detectable threshold was affected by the complex background of soil in practical application, such as the physical or chemical substances that occur naturally in the soil.

The effect of the primer–probe set was evaluated by testing the egg density of field soil and comparing the data obtained using the qPCR, ddPCR, and shallow dish methods. The shallow dish method detected the lowest number of eggs, followed by qPCR, which detected about 1.5-fold more eggs compared to manual counting. The largest number of eggs was detected using ddPCR, representing the most sensitive tool of all three testing methods. It has been proven that the extraction efficiency of soil nematodes using the shallow dish method is generally poor, affected by the hatching rate of eggs and the mobility of nematodes in soil [35]. Furthermore, this method is time-consuming to collect and observe the nematodes, and it needs further confirmation of nematode population. The higher detection efficiency of egg density using qPCR and ddPCR was attributed to the simultaneous detection of both hatching and dormant eggs. As the plant endoparasite, hatched J2s of RKNs are vulnerable to environmental stresses and are viable in the soil for periods much shorter than unhatched eggs [36]. Survival of RKNs between two host crop planting seasons relies on the ability to remain dormant during adverse conditions, when the egg is the main survival stage in soil [37,38]. By detecting the population density of eggs at the beginning of the crop, we can compare the value with the economic threshold of target RKN and implement control strategies in time to hinder the population from attaining the economic injury level [39].

In conclusion, we developed a novel primer–probe set which was species-specific for *M. enterolobii*. It could be applied for identification and quantification of *M. enterolobii* eggs in soil via qPCR and ddPCR platforms, without procedures to isolate a single juvenile or adult. For the first time, ddPCR was applied to quantify PPN species, which exhibited better performance and could provide an alternative strategy of detection. Together, our research can facilitate early diagnostic and accurate quantification of *M. enterolobii* in field. The obtained results are crucial for early evaluation of disease occurrence caused by *M. enterolobii*, which can help growers to adopt targeted and suitable management strategies.

## 4. Materials and Methods

### 4.1. Preparation and Identification of Nematodes Populations

A total of eight species of nematodes, i.e., five *Meloidogyne* spp., namely, *M. enterolobii*, *M. incognita*, *M. javanica*, *M. graminicola*, and *M. arenaria*, two phytoparasitic nematodes (*H. glycines* and *R. reniformis*), and one nonparasitic nematode of *C. elegans*, were used in this study as listed in Table 3. The parasitic nematode *C. elegans* was maintained at 20 °C on nematode growth medium (NGM) agar plates, and the adult females were picked from the petri plates. Single females of other plant parasitic species were excised from host roots under a stereoscopic microscope. These species were identified separately via PCR using species-specific primers [24,40,41,42,43,44]. The eggs of *M. enterolobii* were obtained from galled host roots and cleaned using the centrifugal/sugar flotation technique [45].

### 4.2. DNA Extraction of Nematodes from Pure Culture and Soil

Different methods were adopted to extract DNA from single females, nematode eggs, and soil samples. (i) The DNA extraction method from single females was modified according to Holterman et al. [46]. Female nematodes of each species were handpicked and soaked in 10 μL of sterile water. An equal volume of lysis buffer was added and digested by proteinase K at 65 °C for 90 min, followed by 10 min incubation at 95 °C. The lysates were centrifuged at 12,000 rpm for 5 min, and the supernatant was transferred to a new tube. (ii) The genomic DNA directly from nematode eggs was extracted using the Dneasy Blood and Tissue Kit (QIAGEN Inc., Hilden, Germany) according to the animal tissues extraction protocol, with a final elution volume of 100 μL of ddH_2_O. (iii) The total DNA from 0.25 g soil samples containing nematode eggs was extracted using the TIANamp Soil DNA Kit (Tiangen Biotech Co., Ltd., Beijing, China) following the manufacturer’s instructions, with a final elution volume of 30 μL of ddH_2_O.

The extracted DNA templates were assessed using a Nano Drop™ 2000 spectrophotometer (Thermo Fisher Scientific, Waltham, MA, USA) to confirm the quantity and integrity. All templates were stored at −20 °C until use. If needed, the DNA samples were serially diluted with sterile water before qPCR or ddPCR assays.

### 4.3. Desigh of M. enterolobii-Specific Primers and TaqMan Probe

The rDNA sequences of *Meloidogyne* spp. Were obtained from GenBank Databases according to the accession number: MF467277.1 for *M. enterolobii*, U96304.1 for *M. incognita*, U96303.1 for *M. hapla*, GU432775.1 for *M. minor*, U96305.1 for *M. javanica*, U96301.1 for *M. arenaria*, JX885742.1 for *M. hispanica*, KY660543.1 for *M. graminicola*, AY281853.1 for *M. fallax*, U96302.1 for *M. chitwoodi*, JN241889.1 for *M. naasi*, KR535971.1 for *M. mali*, JN241881.1 for *M. marylandi*, JN241896.1 for *M. graminis*, and KF482367.1 for *M. ethiopica*. These sequences were aligned via CLUSTALW (https://www.genome.jp/tools-bin/clustalw, accessed on 2 September 2021) using default parameters. The primers and TaqMan probe were selected using Primer Premier 5.0 software (Premier Biosoft International, Palo Alto, CA, USA), and the length of the primers or probe was set as 20 ± 2 bp. To determine the specificity of the designed prime–probe set, the sequences of the primers and probe were separately submitted to the Primer-Blast tool in NCBI (https://www.ncbi.nlm.nih.gov/tools/primer-blast/index.cgi, accessed on 2 September 2021). The designed primers and probe were synthesized by BGI Tech Solutions Co., Limited (Beijing Liuhe, Beijing, China). All primers and the probe were purified by PAGE after synthesis.

### 4.4. TaqMan qPCR Assays

The reaction mixtures for qPCR contained 1× *PerfectStart*™ II Probe qPCR Mix (TransGen Biotech, Beijing, China), 0.2 μM each of forward and reverse primers, and 0.2 μM of TaqMan probe. Then, 1 μL of target DNA template was finally added to reach a total volume of 20 μL. All mixtures were measured in triplicate on QuantStudio™ 6 Flex Real-Time PCR System (Applied Biosystems, Foster City, CA, USA). The qPCR program was set as follows: an initial enzyme activation step of 30 s at 94 °C, followed by 40 cycles of 5 s at 94 °C and 1 min at 60 °C. The mean cycle threshold (Ct) values were calculated from three reactions of each triplicated set.

### 4.5. ddPCR Assays

The ddPCR workflow was carried out on a QX200 AutoDG ddPCR System (Bio-Rad Laboratories, Hercules, CA, USA) according to the manufacturer’s instructions. Each ddPCR reaction mixture consisted of 1× ddPCR Supermix for Probes (No dUTP, Bio-Rad Laboratories, Hercules, CA, USA), 0.90 μM of forward and reverse primers, 0.25 μM of TaqMan probe, and 1.1 μL of DNA template, with a total volume of 22 μL. Then, 20 μL of the mixture was used to generate droplets in a Bio-Rad Automated Droplet Generator according to the manufacturer’s manuals. The resulting emulsion was finally transferred into a new 96-well PCR plate, which was heat-sealed with a pierceable foil in a PX1™ PCR Plate Sealer (Bio-Rad, Laboratories, Hercules, CA, USA) at 180 °C for 5 s. The PCR plate was then put into a T100™ Thermal Cycler (Bio-Rad Laboratories, Hercules, CA, USA) to proceed with PCR amplification. The cycling conditions were set as follows: 95 °C for 10 min, 40 cycles of 94 °C for 30 s and Tm for 1 min, and a final step of 10 min at 98 °C. To optimize the annealing temperature, Tm was set ranging from 50 to 70 °C in the gradient block of PCR instrument, and the ramp rate was 2 °C/s. After amplification, the droplets on the plate were read using the Droplet Reader and analyzed using QuantaSoft™ Analysis Pro software (version 1.0.596, Bio-Rad Laboratories, Hercules, CA, USA). Thresholds were manually set for each sample to discriminate positive/negative droplets. The absolute copy number was calculated according to the Poisson distribution, and the droplet counts were converted to copies·μL^−1^. The optimal annealing temperature was selected on the basis of the highest temperature to produce the best separation of negative and positive droplets. All samples were measured in triplicate.

### 4.6. Specificity Tests for Designed Primers and TaqMan Probe

To evaluate the specificity of the designed primer/probe set, eight species of nematodes (listed in Table 3) were selected as the target detection objects for qPCR or ddPCR analysis. For each species, DNA templates were extracted from a single female using the proteinase K digestion method, as described above, and 1 μL DNA solutions were used for further detection. For qPCR analysis, the fluorescence data were collected and plotted versus Ct and dRn to obtain the amplification curves for each species. For ddPCR analysis, the fluorescence amplitude plots for each species were compared.

### 4.7. Standard Curve Determination of Genomic DNA and Total DNA from Soil Containing Nematode Eggs

The gDNA solutions of *M. enterolobii* extracted from eggs were diluted to 5000, 1000, 500, 100, 50, 10, 5, and 1 pg·μL^−1^. Then, 1 μL of these DNA dilutions were tested using qPCR and ddPCR as previously described. In addition, DNA templates isolated from 0.25 g of sterilized soil artificially inoculated with 1, 5, 10, 25, 50, 100, and 200 eggs of *M. enterolobii* were also analyzed. To calculate the amplification efficiency of the designed primer–probe set, twofold dilution series of total DNA extracted from soil containing 200 eggs were prepared. All samples for both PCR assays were amplified in triplicate, and the average obtained DNA quantities, represented as Ct values (qPCR) or copy numbers·μL^−1^ (ddPCR), were used to generate standard curves. For qPCR assays, standard curves were constructed by plotting the Ct values against the logarithm of the DNA quantities (pg) or number of inoculated eggs. For ddPCR analysis, standard curves were determined by plotting the number of quantified DNA copies·μL^−1^ against the DNA quantities (pg) or number of inoculated eggs. The relative standard deviation (RSD = standard deviation/mean × 100%) was also calculated for each sample to assess variability.

### 4.8. Minimum Threshold Detection for qPCR and ddPCR Assays

To compare the detection limit for qPCR and ddPCR assays, serial dilutions, tenfold gradient dilutions (ranging from 1000 to 1 fg·μL^−1^) of gDNA extracted from pure culture or five-, 10-, and 100-fold dilutions of total DNA from 0.25 g of soil containing one egg, were prepared. Then, 1 μL of the samples was tested using both methods as mentioned above. Only the sample giving a Ct value below 38.5 in qPCR analysis or exhibiting more than three positive droplets in ddPCR was considered as positive.

### 4.9. Quantification of M. enterolobii Eggs in Field Samples

Approximately 200 g of fresh soil samples were collected in an eggplant growing field in Wenchang, Hainan. The soil was mixed thoroughly, and 10 g of fresh soil was used to measure soil moisture content via the oven-drying method, in order to convert the weight of tested fresh soil into dry soil weight [47]. Then, 100 g of fresh soil was weighed for further analysis. Three subsamples of 0.25 g soil were taken for total DNA extraction as mentioned above. The remaining soil was placed into a shallow covered dish for hatching. After 3 days, water from the dish was passed through a 500-mesh sieve to collect nematodes. Then, the nematodes were counted under microscope. The total DNA solutions were analyzed using qPCR and ddPCR, and the obtained data were fitted to the corresponding standard curve formula to calculate the number of eggs in 0.25 g of fresh soil. Lastly, the egg density of soil sample was interpreted as number of eggs per gram of dried soil.

### 4.10. Statistical Analysis

Statistical analyses were carried out using the OriginPro 2021 software (OriginLab, Northampton, MA, USA). All data obtained in this study were imported to the Originlab workbook, which had an analysis toolbar to calculate standard deviations, determine the coefficient of determination (*R*^2^), and generate the linear fitting for standard curves.

## Figures and Tables

**Figure 1 ijms-23-11185-f001:**
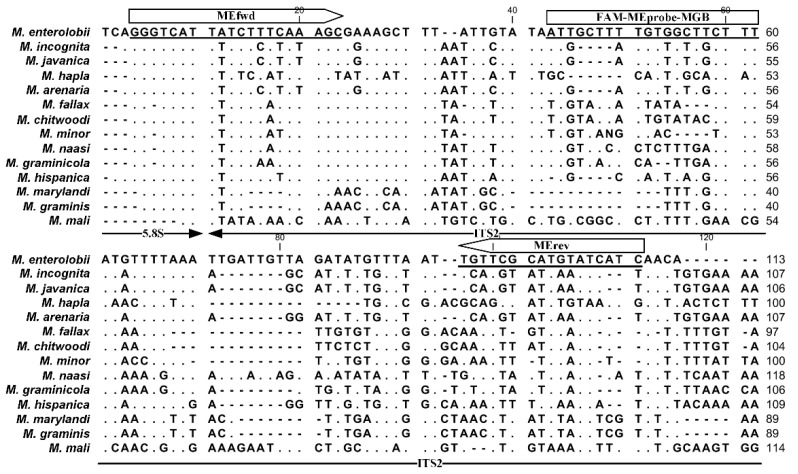
Designed primers and Taqman probe-targeted ITS2 sequence of *M. enterolobii* in this study. The dots and dashes indicate conserved nucleotides and gaps in the sequences, respectively.

**Figure 2 ijms-23-11185-f002:**
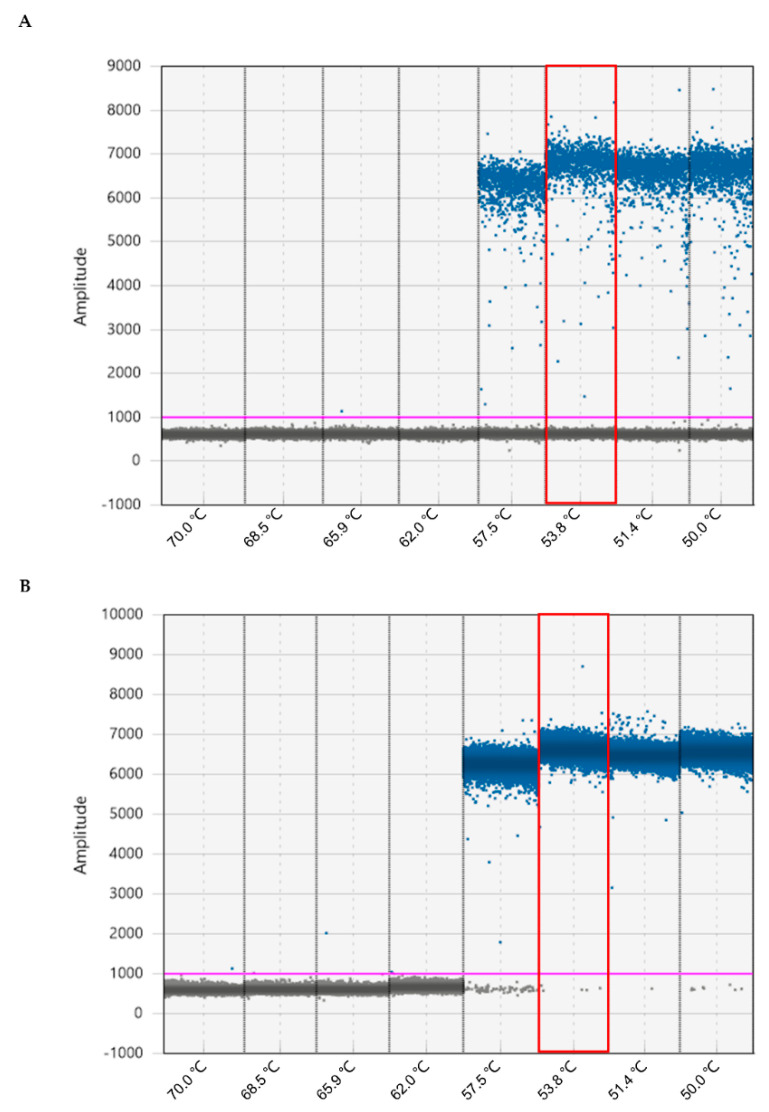
Optimization of annealing temperatures for ddPCR assay to identify *M. enterolobii*. The total DNA extracted from 0.25 g of soil containing 200 eggs (**A**) and 5 ng of genomic DNA extracted from eggs (**B**) was detected under gradient temperature from 50 °C to 70 °C. The droplets achieved at the optimal annealing temperature are indicated in the red box. The blue and gray dots indicate positive and negative droplets. The pink lines represented the horizontal threshold value of 1000.

**Figure 3 ijms-23-11185-f003:**
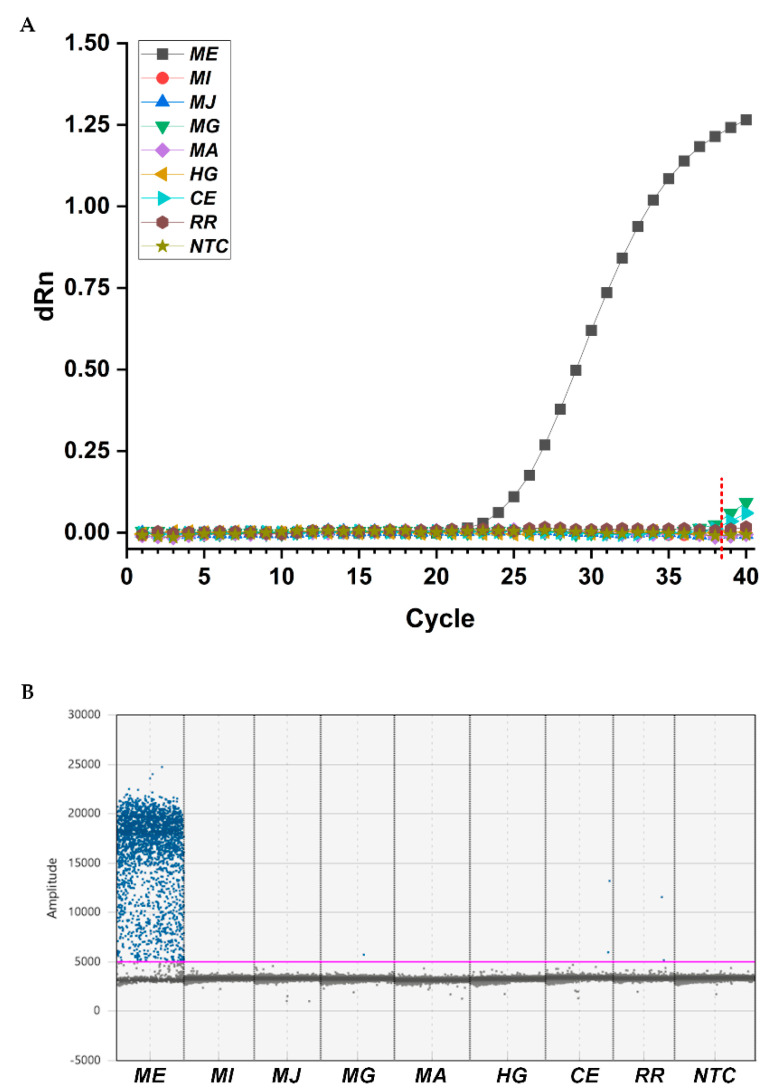
Specificity tests for primers/Taqman probe set analyzed by qPCR (**A**) and ddPCR (**B**) to identify *M. enterolobii*. ME, *M. enterolobii*. MI, *M. incognita*; MJ, *M. javanica*; MG, *M. graminicola*; MA, *M. arenaria*; HG, *H. glycines*; CE, *C. elegans*; RR, *R. reniformis*; NTC, no template control. dRn, normalized fluorescence. The red dashed line indicates the cutoff point in qPCR. The pink line represents the horizontal threshold value of 5000 in ddPCR.

**Figure 4 ijms-23-11185-f004:**
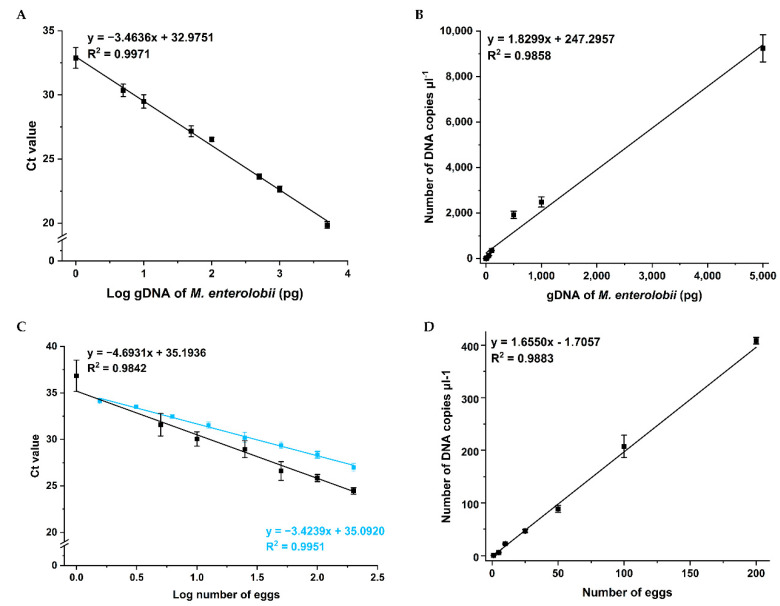
Regression lines for DNA samples of *M. enterolobii* detected by qPCR and ddPCR. (**A**) The standard curve built from Ct values obtained from qPCR against the logarithm of gDNA quantities. (**B**) The standard curve built from number of DNA copies·μL^−1^ obtained from ddPCR against the gDNA quantities. (**C**) The standard curves built from Ct values obtained from qPCR against the logarithm of number of eggs inoculated into 0.25 g of soil. The curves were generated using DNA solutions of the handpicked number of eggs added to the soil (black) or twofold dilution series of DNA solutions extracted from soil containing 200 eggs (blue). (**D**) The standard curve built from the number of DNA copies·μL^−1^ obtained from ddPCR against the number of eggs inoculated into 0.25 g of soil.

**Figure 5 ijms-23-11185-f005:**
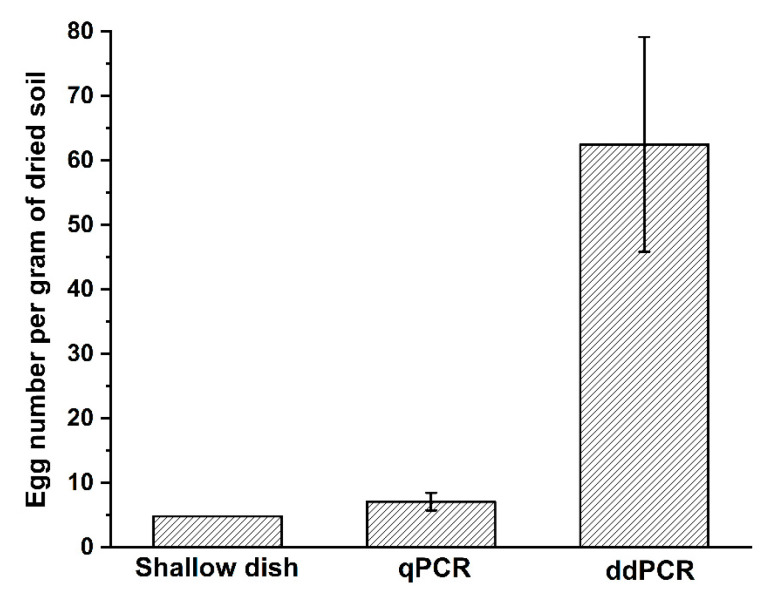
Egg density of *M. enterolobii* in field soil determined using shallow dish, qPCR, and ddPCR methods, respectively. The value obtained using the shallow dish method was calculated as a function of the number of juveniles hatched from soil, which was assumed as the number of eggs in soil.

**Table 1 ijms-23-11185-t001:** The data obtained from qPCR and ddPCR for gDNA dilutions and total DNA templates extracted from eggs in soil.

	qPCR (Ct)		ddPCR (Copies·μL^−1^)			qPCR (Ct)		ddPCR (Copies·μL^−1^)	
gDNA (pg)	Mean ± SD	RSD (%)	Mean ± SD	RSD (%)	Number of Eggs in Soil	Mean ± SD	RSD (%)	Mean ± SD	RSD (%)
1	32.877 ± 1.972	5.997	0.261 ± 0.028	10.728	1	36.839 ± 1.694	4.598	0.284 ± 0.011	3.873
5	30.361 ± 1.187	3.910	5.510 ± 0.210	3.811	5	31.580 ± 1.216	3.851	5.827 ± 0.363	6.230
10	29.489 ± 1.282	4.349	28.300 ± 1.707	6.032	10	30.050 ± 0.771	2.566	22.488 ± 1.252	5.567
50	27.157 ± 1.040	3.830	129.767 ± 8.619	6.642	25	28.945 ± 0.909	3.140	46.633 ± 2.439	5.230
100	26.527 ± 0.412	1.553	358.556 ± 2.169	0.605	50	26.603 ± 1.001	3.763	88.433 ± 6.436	7.278
500	23.634 ± 0.513	2.172	1925.167 ± 159.269	8.273	100	25.836 ± 0.389	1.506	207.333 ± 21.362	10.303
1000	22.651 ± 0.573	2.530	2485.778 ± 222.958	8.969	200	24.450 ±0.347	1.419	408.556 ± 6.345	1.553
5000	19.856 ± 0.656	3.306	9242.889 ± 599.605	6.487					

**Table 2 ijms-23-11185-t002:** The minimum threshold for identification of *M. enterolobii* detected by qPCR and ddPCR methods.

Eggs in Soil	qPCR (Ct)	ddPCR (Copies·μL^−1^)	DNA Concentration (fg·μL^−1^)	qPCR (Ct)	ddPCR (Copies·μL^−1^)
1/30	36.814	0.393	1000	32.901	0.283
1/150	37.748	0.256	100	37.195	0.164
1/300	/ ^1^	0.103	10	/	0.055
1/3000	/	/	1	/	/

^1^ “/” denotes no positive results.

**Table 3 ijms-23-11185-t003:** The sources of *Meloidogyne* spp. and other nematodes selected in this study.

Species	Origin	Host
*M. enterolobii*	Wenchang, Hainan	Pepper
*M. incognita*	Zhengzhou, Henan	Tobacco
*M. graminicola*	Haikou, Hainan	Rice
*M. arenaria*	Langfang, Hebei	Tomato
*M. javanica*	Chengmai, Hainan	Tomato
*C. elegans*	Haikou, Hainan	/
*H. glycines*	Langfang, Hebei	Soybean
*R. reniformis*	Chengmai, Hainan	Banana

## Data Availability

Not applicable.

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
