# Peer review of "Development of a Novel Primer–TaqMan Probe Set for Diagnosis and Quantification of Meloidogyne enterolobii in Soil Using qPCR and Droplet Digital PCR Assays"

_ijms, 2022, doi:10.3390/ijms231911185_

Round 1

Reviewer 1 Report

The manuscript of Y. Chen et al. reports methodological developments for the detection and quantification of Meloidogyne enterolobii, a plant parasitic nematode. The pathogen causes root-knot, an economically important disease of a broad range of cultivated crops.

The work is of interest, and the developed methods are important. In general, the method development work was conducted correctly. The major fault I think is the handling of qPCR Ct (Cq) data for the lower DNA amounts. First, a qPCR to be used for diagnosis and/or quantification, a cut-off, a Cq value discriminating positive and negative samples is needed. Then, using the cut-off, the limit of detection can be determined. It seems (see Figure 3) that even for non-M. enterolobii samples, the qPCR results in amplification in later cycles (38-40), and therefore, results in Ct value. This is common for qPCR, and not an error by the authors, but should be addressed in the work.

Second thing is, the qPCR efficiency of samples originating from M. enterolobii eggs in soil. According to Figure 4, the slope of the curve is -4.6931, which points to an efficiency of ~63.3% which is quite low (and not stated in the manuscript). Calculated from the dilution series of pure DNA extracts the efficiency was ~94.8% which is acceptable. But, in this case, for 50 pg DNA the resulting Ct was 26.362, and for 100 pg (the double amount) was 26.163, which is almost the same, but at efficiency close to 100%, the Ct difference should be 1.

For 1 egg in the soil, the qPCR resulted in Ct = 36.839 (Table 1), and for the second experiment, for the determination of the minimum detection treshold, 1/30 eggs resulted in Ct = 36.814 (Table 2), which is almost the same for 30x DNA amount differences. This can be caused by inconsistencies in the efficiency, poor repeatability of qPCR and/or problems with unspecific amplification. The reason can be any of these, please investigate carefully, and describe.

I suggest the authors to provide more repeats on the dilution series, and the calculated efficiencies. It should be carefully addressed which DNA dilutions to use / to accept as belonging to the log-linear range of qPCR.

Based on my judgement, the digital droplet PCR results and the method development can be accepted.

I feel that the language/English of the manuscript needs a bit of polishing.

There are several minor points that I would like the authors to consider in order to improve the manuscript. These I have marked in the PDF version of the manuscript.

Altogether, I suggest major revisions to be done on the manuscript before publication.

Reviewer 2 Report

The authors developed a new primer-probe set which was species-specified for Meloidogyne enterolobii. It could be applied for identification and quantification of M. enterolobii eggs in soil via qPCR and ddPCR platforms, without procedures of isolating single juvenile or adult. However, it cannot be accepted for publication by reason of the following aspects requiring circumstantial revision.

1.       Authors designed a new primer-probe set for detecting Meloidogyne enterolobii. But in systems from complex DNA sources (such as soil containing a wide variety of organisms), How to ensure the specificity of primers?

2.       At present, qPCR is still a relatively reliable quantitative technology. However, the data in Section 2.4 (Table 1), the reliability of qPCR cannot be seen, while the newly developed ddPCR has good quantitative analysis results. Will the authors deny the reliability of qPCR?

3.       In section 2.6, the results of qPCR and shallow dish are basically the same, but the result of ddPCR is very different from those results. How should the authors explain this difference? Which result is closer to the truth?

Reviewer 3 Report

The current "Development of a novel primer-TaqMan probe set for diagnosis and quantification of Meloidogyne enterolobii in soil by qPCR and droplet digital PCR assays" is an interesting read in field of development of molecular markers for quick identification of plant-parasitic nematodes. Several works have been published in the recent past that have proposed many markers and current study makes use of ITS region.

I have few recommendations to further improve the quality of manuscript and few queries that I would like to pose to the authors.

1. The initial part of the manuscript has several grammatical errors and the language could be improved. For example Line 36 is in contradiction to the tense used in the previous sentence. It should have been "The infective second-stage juveniles (J2s) hatch" There are other instances through the introduction such as in Line 50 and Line 87. The discussion, methods are much better. Please revise the manuscript for (few) grammatical errors.

2. My query is with the DNA isolation from soil. Did the soil predominantly contain only eggs? 

3. Line 387. Are the authors confident that the eggs survive the oven-drying that was carried out? Because in the following sentences the same samples were used to hatch for obtaining J2. 

Round 2

Reviewer 1 Report

The manuscript of Y. Chen et al. reporting methodological developments for the detection and quantification of Meloidogyne enterolobii  (ijms-1898679) has significantly been improved during the review process.

The authors set a cut-off, as requested, and was used consequently thereafter. It was done by selecting an artificial value as a cut-off, which is not the best approach, but very much acceptable.

The authors also repeated the dilution series, as suggested, and with the results, most the concerns raised are dispelled. I think that the new standard curve generated during the review process, and the description of testing the 2-fold dilution series of DNA solutions extracted from soil containing 200 eggs of M. enterolobii, should be incorporated in the manuscript text.

In addition, I feel that Response 3 is not convincing (or maybe the authors misunderstood my point). I feel that the Ct~36.8 obtained for 1 egg and also for 1/30 egg DNA equivalents is not an error, but it results from the differences between the two methods, namely, measured with “pure” DNA extracts and measured using DNA from soil, which contains lots of background DNA and also, potentially, PCR inhibitors. This should be addressed in the discussion.

Suggestions for two minor corrections:

Line 20: “and this work presents the first application of this technology for plant parasitic nematodes “instead of “and this technology which is the first attempt application for plant parasitic nematodes.”

L176: “strong”, instead of “strength”

I judge that after considering my suggestions, the manuscript should be accepted for publication.

Reviewer 2 Report

This is an interesting and important new approach. However, high sensitivity will mean that there may be a large number of false positives.

So I suggest the authors provide a simple experiment. After soil collection from nematode-free fields, 10, 50 and 100 eggs of M. enterolobii were added. How many eggs can be determined using established methods qPCR and ddPCR respectively?
